# Finding the True Number of Females with Autistic Spectrum Disorder by Estimating the Biases in Initial Recognition and Clinical Diagnosis

**DOI:** 10.3390/children9020272

**Published:** 2022-02-17

**Authors:** Robert McCrossin

**Affiliations:** 1Cooroy Family Practice, Cooroy 4563, Australia; bobmccrossin@gmail.com; 2Beerwah Family Clinic, Beerwah 4519, Australia

**Keywords:** autistic spectrum disorder, male-to-female ratio, biases, young women

## Abstract

The proportion of females whose ASD diagnosis is missed is unknown. The ratio of males to females with ASD is generally quoted as 4:1, though it is believed that there are biases preventing females from being diagnosed and that the true ratio is lower. These biases have not been clearly identified or quantified. Starting with a clinical dataset of 1711 children <18 years old, four different methods were employed in an inductive study to identify and quantify the biases and calculate the proportion of females missed. A mathematical model was constructed to compare the findings with current published data. The true male-to-female ratio appears to be 3:4. Eighty percent of females remain undiagnosed at age 18, which has serious consequences for the mental health of young women.

## 1. Introduction

### 1.1. Problem Statement

The problem is clearly stated by a 20-year-old from a recent qualitative study of female camouflage [1]:

“The amount of girls that aren’t diagnosed because they are more likely to camouflage than boys is really bad. I went for so long without being diagnosed because they didn’t know that I could pretend to be normal!”

The problem to be solved is the true number of girls with autistic spectrum disorder (ASD). The problem needs to be solved because it is becoming very clear that a large number of girls are being missed, with long-term severe negative consequences affecting mental health, self-perception and access to support [1]. The solution lies in categorising the data of 1711 children with ASD managed in my clinic with the information provided in the detailed histories from the children and their parents. 

### 1.2. The Structure of the Study

The problem is not amenable to solution by the usual deductive methods. This study was inductive [1] and used qualitative and quantitative information. The key step was to reject all speculation, especially information we know is not true, such as current values of the male-to-female ratio, and use only what is definitely known from observation. This must be primarily what the patient or family is telling us. The seminal observation was that the carer’s second child is easier to recognise than the first. The first child diagnosed is equally likely to have siblings of either sex and those with ASD will be distributed in the true gender ratio. There is no prior hypothesis other than that the numerical solution should lie somewhere in the data. The method is analogous to diagnosing a patient but the “patient” is a population rather than an individual. Like any clinical diagnosis, the most important clues are going to be in a detailed history. The process began by seeking patterns in the histories and enabling the children to be categorised into groups such that the problem can be addressed. 

Once patterns became evident, it was realised that clues were already present in published data and these were interpreted to triangulate the study results. Like a clinical diagnosis, this study used cumulative qualitative and quantitative information and is necessarily Bayesian in philosophy [2]. With multiple steps and the law of large numbers operating on a large dataset, an accurate approximation should be possible if the assumptions are valid. Intermediate hypotheses were generated and the supporting information for them was derived from the clinical database, and descriptive and hypothesis testing statistics were applied as needed to validate results. The statistics sources are listed in Appendix E. The outcomes of interest were derived from the database alone and/or in combination with data from published studies, and equations were derived to validate these results.

### 1.3. Study Environment

My paediatric practice is a private clinic focused on behaviour and is based in South East Queensland. It is bulk-billed under Medicare (service free at the point of delivery) and primarily serves the Sunshine Coast and adjacent areas, with a drainage population of about 800,000. It covers the full range of socioeconomic status groups and is likely representative of the Australian population as a whole.

The study group was a total of 1711 children 1–18 years of age with ASD diagnosed and/or managed by me between October 2014 and April 2020. Diagnoses were made using the DSM-5 clinical criteria. The study was stopped when procedural changes in Queensland and Australia, combined with the COVID-19 pandemic, made it likely the population was no longer representative (detailed in Methods 2.14. Aftermath).

In Queensland at that time, the formal diagnosis was made by a specialist paediatrician or child psychiatrist, with or without advice from allied health professionals (AHPs), but commonly assisted by a psychologist in particular. In specialist private practice, all patients had been referred by a general practitioner (family physician) and may have also been seen by AHPs and often provisionally diagnosed prior to referral. An AHP assisted in diagnosing half of my patients. With at least one gatekeeper (GP) and often another (AHP), the important DSM-5 criterion that there was a clinical problem was satisfied.

## 2. Methods, Definitions and Intermediate Results

### 2.1. Male/Female Odds Ratio: Definitions

The male/female odds ratio (MFOR) may be defined in various ways. It is the male-to-female ratio among those with ASD, controlling for the male-to-female ratio among those in the population of interest without ASD [3]. If we start with a population (ASD and not ASD) of equal numbers of males and females the MFOR is the probability of diagnosing a male with ASD divided by the probability of diagnosing a female. It measures the relative prevalence of boys and girls with ASD in their respective populations. It is a ratio of proportions and, mathematically speaking, an example of odds. At birth and during the age range of this study (1–18 years), there is a population excess of males of around 5%. The values for subsets of this age range were found from Australian Bureau of Statistics data from the 2016 census, and the number of internal siblings (the key category for the MFOR calculation and defined below) in each subset was counted and the weighted average male/female ratio was found to be 1.055. This correction was applied as appropriate to the unadjusted case numbers. An MFOR using unadjusted case numbers was designated as uMFOR. The most commonly reported MFOR is around 4:1 [4] (p. 57), though it is believed it is probably closer to 3:1 [3]. It is believed there are remaining biases against females, though these have not been quantified or even clearly defined. Gender fluidity was dealt with by using the sex of assignation at the time of diagnosis. There were only a handful of internal sibling cases and they flowed both ways.

### 2.2. Defining the Biases and Patient Categories

Initially, the family knows there is something odd going on with their child’s behaviour but has no prior experience to explain it. The behaviour is evident in family settings and/or school. After a variable number of iterations with various professionals, a diagnosis of ASD is eventually made. After the diagnosis is made, caregiver knowledge of what to look for usually increases rapidly. 

Two biases will be considered. The first is that of recognition, which is the set of factors preventing the girl from reaching the door of the diagnostician’s clinic. The second is that of diagnosis, which is the set of factors after that point preventing the definitive diagnosis from being made. A bias was quantified as the ratio of an MFOR with the bias to the corresponding MFOR without it. Each bias will have numerous subsets. These were beyond the scope of this study, but this study should set a platform for their elucidation.

The study assumed that the MFOR in simplex families (one child with ASD) would be about the same as for the first child diagnosed in multiplex families (two or more children with ASD). The first child diagnosed in a multiplex family was designated the proband. When simplex and multiplex proband cases were combined or interchangeable, they were termed singletons. Subsequent children in multiplex families were designated siblings and would have a lower MFOR due to increased caregiver awareness. Crucially, these children are not necessarily younger siblings because older siblings may have been missed. It is the order of diagnosis that is critical. Half siblings, step-siblings or foster siblings in the same household as the proband when they were diagnosed were counted since the key factor was increased awareness after proband diagnosis, not genetic relatedness. There were then three family categories: simplex, proband and sibling. The ratio of a singleton MFOR and sibling MFOR in the family diagnosis category (Figure 1) was defined as the recognition bias Br. 

All subjects were either diagnosed by me or, if already diagnosed when referred for management, were assessed by me as meeting DSM-5 criteria. There is variation between clinicians in how these criteria are interpreted; therefore, the diagnostic variability in finding the unbiased MFOR was minimised by not including siblings not personally diagnosed in my clinic. There were two diagnostic categories: (1) Those designated via an external diagnosis were referred already diagnosed, which was normally done formally by a paediatrician or child psychiatrist and corroborated by me during the first consultation. (2) Those referred for confirmation of suspected ASD, a behaviour problem or other diagnosis who on my assessment met DSM-5 criteria for ASD were designated as an internal diagnosis. A diagnosis bias Bd was defined as the ratio of the MFORs of an external and internal family category (Figure 1). If there was residual bias in my diagnosis of girls in either direction, by definition, it was beyond my awareness but the internal cases were the only baseline available to me and the external validity of this categorisation will be explored in the Summation section (Section 2.15).

During the study period, when all cases were diagnosed by a paediatrician or child psychiatrist, the medical clinician was on the watershed, where it was possible to categorise all patients to determine the biases from suspicion to the final diagnosis. The product of recognition bias and diagnosis bias was defined as ascertainment bias. It will be necessary to assess whether my procedure was externally valid, but at least it meant the children were assessed in a consistent manner by a single clinician. The first aim of this study was to derive the unbiased MFOR from the dataset and validate it externally. The sibling data are shown in Table 1. The categories are shown in Figure 1 and the internal sibling category was examined in detail. 

### 2.3. Male/Female Odds Ratio: Results

The only category with the minimum bias was the internal siblings such that the unbiased male/female odds ratio (MFOR) was then 200/(253 × 1.055), i.e., 0.7493 or 3:4. The statistical confidence boundaries are as follows: the 95% confidence interval was 0.620–0.900 and the 99% confidence interval was 0.584–0.954; there was a 95% probability the MFOR was <0.874 and a 99% probability the MFOR was <0.932.

This MFOR is much lower than current estimates and we must rule out possible biases in the methodology, explain any data anomalies and seek validation from independent data. First, there is the comparison with the external siblings. The external MFOR was 1.069 and the external/internal MFOR ratio was 1.426, which measures the diagnosis bias between the groups. The external sibling case numbers were not large, but there was a 99% probability that the external MFOR was <1.92 compared with the current suggested lower limit of 3 [3]. The external and internal ratios were not significantly different: χ^2^ for comparison of proportions gave *p* = 0.1366. This was an issue for calculating the diagnosis bias, which is discussed later. It does show that my diagnostic process was not very different from my medical peers and gives very different results to the currently accepted values. Comparison with my AHP colleagues for the diagnosis of internal siblings appeared to make me an outlier at MFOR 0.709, but when compared with the MFOR of 0.725 for the group, where 21 different AHPs had made the provisional diagnosis, there was little difference. χ^2^ for comparison of proportions gave *p* = 0.9113. The diagnostic pathway was examined in detail.

### 2.4. Internal Sibling Diagnostic Pathway

My practice was to make the diagnosis myself if sure but refer the younger children to a psychologist for assessment if unsure. The third category had already been assessed by an AHP when referred to me. The referral patterns for males and females <3, 3 + 4 and 5 + 6 years of age (Table 2) were examined. 

My clinic did seek diagnostic help for the youngest girls, but this pattern was being repeated by all carers or clinicians, which involved getting AHP testing before medical referral. The MFOR for <7-year-olds was 0.853, reflecting the fact that boys are recognised earlier, and the MFOR for my diagnosis alone for <7-year-olds was 0.874 compared with the AHP-assisted MFOR of 0.835, showing that my assessment was not biased towards the younger girls compared with assessments by or with AHPs. The median age of diagnosis for girls was 7y1m and 6y7m for boys (Appendix D) and the overall MFOR for siblings 7 years and over was 0.665. The conclusions were that all assessors were more comfortable diagnosing girls as they got older, the proportion of girls diagnosed was higher and my clinic was seeing proportionately more girls directly. My apparent diagnostic biases were functions of age-related referral patterns and not a significantly different diagnostic practice.

### 2.5. Were the Sibling Genders Found Truly Random?

The key assumption was that the siblings would be recognised and diagnosed according to the uMFOR. There is one specific situation where this might not be true. If there are two gender-discordant children, both undiagnosed, there is going to be a bias towards diagnosing the boy first. The particular order of concern is if the younger child is male and diagnosed first then there is likely to be an excess of male probands and female siblings due to recognition bias in those designated siblings. It was uncertain how to interpret a result if the proband was external because of the effect of diagnosis bias. The situation where the first diagnosed was younger than the second in sibships of any size, where the proband and first sibling were internal and gender discordant was examined. There were 11 younger female probands (38% of total) and 24 younger male probands (35% of total), giving 13 excess female siblings. The difference was not a surprise, but the fact that it is as common for female probands to have older brothers as the reverse was not expected.

When we look at a cross-sectional sample of the population at different ages, as we are here for ASD siblings, and adjust for excess males, it is a proxy for an assumed cohort, where we start with equal gender numbers and assume that with no bias, boys and girls with ASD were equally likely to be diagnosed by an arbitrary cut-off, here 18 years. There was, however, a lag at the older end because, in this group, there was a delay in diagnosing girls. The number of girls in the cohort missed after age 18 years compared with the boys could be estimated. The 90th centile for diagnosing girls was 14y5m, 1y4m after the boys (Appendix D). Over the last 3 years (15–18), 9, 7 and 8 girls were diagnosed, averaging 8/year; therefore, we can say that about 8 × 4/3 = 10.7 extra girls should be counted. The difference between excess girls found in the sibling dyad and cohort girls missed was two girls. There were also the girls who were never found since there was likely to be incomplete recognition of an uncertain magnitude before 18 years. 

Another way of assessing whether the unknowns cancel, in particular, male bias in the initial younger brother/older sister dyad, is to compare the MFOR of siblings in families where there is a proband/sibling pair only with the MFOR of the balance of the internal siblings from larger families where the recognition should be greater. These were 117/(148 × 1.055) and 83/(105 × 1.055), giving 0.7493 and 0.7493, respectively. This was a numerical fluke but does suggest the overall MFOR result of 0.7493 is a reliable estimate.

### 2.6. Translating the MFOR to Numbers of Practical Importance

Finding the MFOR does not directly lead to the true female prevalence or estimate the proportion of girls missed. These can be derived from published data, combined with the MFOR or using Bayes’ theorem where the true ASD prevalence can be found using a more visible comorbidity as a tag. If we know the proportion of the tag in ASD and, conversely, the proportion of ASD in the tag plus the prevalence of the tag condition, we can use Bayes’ theorem to find the prevalence of ASD. Bayes’ theorem is integral to clinical diagnosis, though we often do not realise we are using it [5]. There is an argument about the precise definition of many psychiatric conditions but this does not invalidate the calculation if the conditional probabilities employ similar definitions. 

The conditional probabilities and prevalence are available for borderline personality disorder (BPD). Data are available for the lifetime prevalence of BPD in women, i.e., 6.2% [6], and adult data, which should reflect lifetime prevalence for BPD in ASD, i.e., 6 of 40 = 15% [7], and ASD in BPD, i.e., 6 of 41 = 14.6% [8]. ASD is, of course, a lifelong condition. The conditional datasets are unfortunately small but they are all we have and we shall see where they lead. By Bayes’ theorem:
P(ASD|BPD) × P(BPD) = P(BPD|ASD) × P(ASD)0.146 × 0.062 = 0.15 × P(ASD)P(ASD) = 0.060


If we know that the proportional of BPD in the general population is 0.062, the proportion of BPD in ASD is 0.15 and the proportion of ASD in BPD is 0.146, then we find that the proportion of ASD in the general population is 0.060.

This indicates 6.0% = 1/16.7 women have ASD (Figure 2). A recent US study [9] provides valuable data to use in the calculations. ASD diagnosis was done using the DSM-5, the age range was similar (3–17 years), data was collected at about the same time (2014–16) and it is the most recent comparable peer-reviewed information. The female prevalence estimate of 1.25% was 1/80 and 1/16.7 was 4.8/80. If we find one girl and miss 3.8 we are missing 3.8/4.8 × 100% or 79% of the girls. The same study showed an overall ASD prevalence in the United States from 2014 to 2016 of 2.47%. There was no statistical increase over that time, though the number was increasing slightly. This may be a true number change. As knowledge of ASD has increased, the prevalence will have followed a sigmoid learning curve, and while it is certainly flattening, the asymptote has probably not yet been reached (shown in stylised form in Figure 3).

This asymptotic approach has different implications for each gender. For males, it suggests that the limit of detection by DSM-5 is being approached. For females, it suggests that no major advance in reducing bias has occurred during the study period.

The overall prevalence in 2016 was 2.76%. The male prevalence over the 3 years was 3.63%; therefore, the likely male prevalence in 2016 would be (2.76 × 3.63)/2.47 = 4.06% and it may have a little way to go to truly flatten with diagnostic saturation. Applying the MFOR of 0.7493:1 the corresponding female prevalence would be 4.06/0.7493%, i.e., 5.42% or 1 in 18.5 girls. This translates to missing 77% of the girls.

The ASD prevalence estimates are strikingly divergent from current estimates, but an independent tag and published data have given similar results congruent with my finding. The result using my data is derived from the MFOR and is relative to the diagnostic rate for males, but the Bayesian result is entirely independent and depends only on data for females.

We have now derived the male/female odds ratio (MFOR) and, combined with published data, have found that nearly 80% of girls with ASD are missed. We will now use the entire patient database alone to estimate the biases and the proportion of girls missed and derive an equation to model the current published MFOR values. We will also use independent information from the patient histories to corroborate the results. 

### 2.7. The Algebra of the Biases

The variables in the final formulae are expressed as probabilities (proportions), odds ratios and biases. The notations used are outlined in Table 3.

The reciprocal of the bias gives the proportion of girls found when the bias operates. It is compared with the males where it is assumed that there is no bias and the two groups are starting with equal numbers, having adjusted for the male excess in the general population. Whether we are finding 100% of the males is moot (Figure 3), but it must form the baseline for gender comparison. Then, if 100% of males are found and the bias is 5, 20% of the females are found and 80% are missed. Crucially, recognition bias must always precede diagnosis bias in the ascertainment process.

If the recognition bias Br is present, the proportion of girls found is 1/Br and the proportion missed is 1–1/Br. It is this proportion 1/Br who will be assessed and subject to diagnosis bias Bd. The proportion finally ascertained with ASD is 1/BrBd. BrBd is defined as the ascertainment bias Ba. The population proportion missed on diagnosis is 1/Br−1/BrBd, which reduces to (Bd−1)/BrBd or (Bd−1)/Ba. The effect of Bd is always going to be less than Br for a given value since Bd operates on proportion 1/Br of the population and 1/Br is always <1. If Br is large, the effect of Bd will be small on the population as a whole since the function is hyperbolic. This is shown in Figure 4, where 80% of girls are missed if Ba = 5 and let Br = Bd = 5^0.5^. 

If only 45% of girls are recognised, then 45% of the 45% i.e., 20% are finally diagnosed. A large recognition bias will make losses on diagnosis small at a population level. The proportions found rapidly drop at first, but as the slope flattens, they become relatively insensitive to minor changes in the bias. The importance of diagnosis bias depends on your perspective. If you are a girl about to be tested, it is very important. If you are a planner deciding where to put funding, then improving recognition may well be more important.

### 2.8. Diagnosis Biases

There were two in the dataset. The first was the singleton (proband + simplex) bias, which is the ratio of the external singleton unadjusted MFOR (uMFOR) to the internal singleton uMFOR. The other was the sibling bias, which is the ratio of the external sibling uMFOR to the internal sibling uMFOR. For the biases, the factor of 1.055 cancelled and the actual numbers of cases can be used. Neither bias was statistically significant when using χ^2^ for the comparison of proportions; therefore, a meta-analysis was done using Stouffer’s method [10], where the sum of the Z-scores between the external/internal ratios, giving the singleton and sibling biases, was divided by 2^0.5^. Without this procedure, it could be assumed that there was no significant diagnosis bias at all and this would be a serious type II error. Z for the singleton ratios was 1.3327 and 1.4732 for the sibling ratios. The combined Z was 1.985, giving a *p*-value of 0.0236. During the data collection, the external MFORs were always greater than the internal ones and we are interested principally in one tail but the two-tailed *p* would be 0.0472. The two biases were then combined using a weighted mean into the diagnosis bias Bd:
[(∑singletons × ext sing ratio/int sing ratio) + (∑siblings × ext sib ratio/int sib ratio)]/∑ singletons + siblings


An important inference from the fact that both diagnosis biases were small and similar is that the external sibling MFOR after removing the weighted mean diagnosis bias was little different from the key variable, i.e., the internal sibling MFOR; it was 0.845 compared with 0.749, while χ^2^ for the comparison of proportions gave *p* = 0.612. This implied that siblings were presenting to my medical peers and myself in much the same gender proportion. 

### 2.9. Correction to an MFOR with Recognition Bias Br Only

In order to make the true MFOR (designated Ro) as accurate as possible, only children consulted in my clinic were counted as probands or simplex. If there were one or more prior diagnosed siblings with ASD, the proband or simplex gender ratio would be falsely low because a proportion Pb will have prior diagnosed siblings not evaluated by me and will themselves be siblings without recognition bias. The ratio of external siblings to external singletons was used as an estimate for Pb. The ratio of external siblings to external singletons should approximate the proportion of internal singletons who are actually siblings themselves of external probands. An external singleton is about as likely to have an older sibling as a younger one; therefore, the latter can serve as an estimate of the former. Possible differences between simplex and proband were neglected because any numerical effect on the final estimates was likely to be small. The value for Pb was 0.227, where this value derived from externally diagnosed patients was deemed appropriate when estimating recognition bias and the biased MFOR because the external data were generalisable in calculating both these results, which reflected external assessments. 

The internal proband ratio was first used for calculating Br because it related directly to the siblings. A case can be made for including the simplex cases as well since there is no obvious biological difference to the probands and, therefore, the weighted mean of both groups (internal singleton MFOR) was used for a separate estimate of Br. This is discussed in detail later. 

To derive a value for Br from the probands, the internally diagnosed proband MFOR (Rpr) was then corrected for the unrecognised prior siblings and the ratio (corrected MFOR of internally diagnosed probands with Br)/(MFOR with no Br) was the true recognition bias Br, i.e., Br = corrected Rpr/Ro

The derivation of the corrected Rpr is shown in Appendix A. The formula is
(Rpr − Ro.F)/(1 − F) where F = Pb.Pf.(Rpr + 1)


### 2.10. External Validity of the Variables

Do the variables derived from my data reflect published results? To derive an MFOR including recognition and diagnosis biases, we start with a population of ASD children distributed in the proportions of the unbiased MFOR as Ro/1 males to females. This is the algebraic equivalent of the proportion of males Pm to the proportion of females Pf. The population has siblings (broadly defined as with minimum recognition bias) in proportion Pb as derived above. The siblings will be distributed by gender in proportions Pm and Pf. Female siblings of male singletons will be recognised according to the diagnosis bias rate Bd. Let its reciprocal be Bdi. Only the proportion Bai of the female singletons will be ascertained as having ASD, where Bai is the reciprocal of the ascertainment bias Ba. Their male siblings will be ascertained but their female siblings will be subject to diagnosis bias Bd. The major proportion of the female singletons (1-Bai) will not be ascertained due to ascertainment bias Ba. Their male siblings will be ascertained but their female siblings will be subject to Ba because the female singletons themselves were not ascertained. All the variables in the equation were derived from the unbiased MFOR Ro, the recognition bias Br, the diagnosis bias Bd and the ratio of external siblings to external singletons Pb. These in turn were all derived from the study dataset. The biased MFOR was then the sum of all the boys divided by the sum of all the girls. In Appendix B, the groups outlined above are shown in Table A1 and the derivation of the biased MFOR is given. The formula is
Ba.Ro(Pb + 1)/(1 + Pb.Pf(Ro.Bdi.Ba + Bdi − Bai + 1))


From this, the outcome of the biases in the ASD population can be calculated using the complete dataset of ASD children in Table 4.

The calculation for the multiplex proband + sibling categories (detailed in Appendix B) gave:

Biased MFOR = 3.31. Percentage of singleton females missed: recognition—79.1, diagnosis—4.3 and total—83.4.

For every 100 singleton boys found, we should find 133 girls. We found 22 and missed 111.

For the weighted mean of simplex and proband categories, the corresponding results were:

Biased MFOR = 2.76. Percentage of singleton females missed: recognition—73.1, diagnosis—5.7 and total—78.8. 

For every 100 singleton boys found, we should find 133 girls. We found 28 and missed 105. 

A recent meta-analysis [3] found that good-quality clinical studies yielded a mean MFOR of 3.32 and they concluded the likely value was closer to 3. My results are consistent with these findings. 

### 2.11. Which Internal Singleton MFOR?

It is not obvious which internal singleton MFOR is the better one to use to calculate the recognition bias. The proband value relates to the actual siblings, but assuming that the simplex children were not fundamentally different, the weighted mean of proband and simplex is derived from a larger number of cases. The recognition bias was much larger than the diagnosis bias and must precede it in the calculation and the results are where the slope of the hyperbolic function is flattening and, therefore, are not very sensitive to variation. For clinical and policy simplicity, Br was rounded to 4 and Ba to 5, giving Bd a value of 1.25 (see the Summation section (Section 2.15) for details). The MFOR can be rounded to 0.75; therefore, we then had four girls for every three boys. We had 75% of girls not recognised, one in five of the 25% assessed were not diagnosed, leading to 80% overall being missed (Figure 5). If we used these variables in the biased MFOR equation, together with a Pb of 0.227, the only variable not derived from Br, Bd and the unbiased MFOR, we found a biased MFOR of 2.88, and for every 100 singleton boys found, we missed 107 girls. For comparison, the MFOR of the weighted prevalences of 3.63% for males and 1.25% for females in the recent US study [9] is 2.90. This derivation is shown graphically in Appendix C. The close agreement suggests the bias model and study values were accurate and generalisable.

The finding that 20% of assessed girls are missed is highly clinically significant and shows the importance of the diagnosis bias meta-analysis, but this is only 5% of the total girls with ASD in the population. Whichever set of these biases is chosen, there are overall more singleton girls who are missed than boys who are found. The effect of the final biases on the proportion of girls found is shown in Figure 5.

### 2.12. Explaining a Data Anomaly

A puzzling finding in the database was that the internal simplex MFOR was persistently about 0.5 less than the internal proband MFOR. This was not statistically significant using χ^2^ for comparison of proportions, but it was stable and the issue was whether the non-significance was a type II error and the data were trying to say something. It would require something different happening in each group and one possibility was that different proportions of the singletons were in fact siblings. The dataset records detailed family relationships. The singletons will be distributed by gender by MFOR (Ro), i.e., Ro/1. We assume that all the males are found. Br is the recognition bias and Pb is the external sibling/singleton ratio describing the proportion of singletons who are actually siblings and do not have recognition bias.

Simplex cases may have at least one prior diagnosed sibling not managed by me. Probands have at least one sibling diagnosed after them managed by me; therefore, to have at least one prior diagnosed sibling, they must have at least two siblings and, therefore, the proportion must be less than Pb because fewer families have larger numbers of children. The total number of families with a proband and one internal sibling (206) and the number with a proband and at least two internal siblings (120) were counted. Then the correction factor for Pb for the proband category will be the proportion of families with one or more siblings who have more than one sibling, which was 120/326, i.e., 0.368.

The proportion of females recognised is 1 × 1/Br. The proportion who are siblings is Pb × 1/Br. However, this is not relevant in this group because they are included in the proportion 1/Br who have all been found. In the group not recognised (1–1/Br), there are Pb (1–1/Br) siblings who, as siblings, will be found. The total proportion of females found is then 1/Br + Pb(1–1/Br) and the singleton MFOR is Ro/[1/Br + Pb(1–1/Br)], which simplifies to
RoBr/[Pb(Br − 1) + 1]


The proband MFOR formula is then
RoBr/[0.368Pb(Br − 1) + 1]


If we used the simplified Br of 4 and MFOR of 0.75 in the singleton equation, we found that MFOR = 1.785, with the actual value being 1.979. In the proband equation, we found MFOR = 2.399, with the actual value being 2.349. The first derived value was 10.9% too low, the second one being 2.1% too high. The reasonably close match of model and data implied sibling recognition accounted for most of the anomaly. 

### 2.13. Corroboration of the Recognition Bias

Asking about camouflaging is part of my standard clinical history and the experience of the families of 100 school-age diagnosed girls in consecutive clinics in my practice was sought. The criterion for camouflage used was being Ms Jekyll at school and Ms Hyde at home, with the transition at the school gate:
“I fall to pieces.”[1]
“I was unbearable with my mother, but at school I was perfect.”[11]


The study was effectively random, had a 100% response and 88 girls behaved in this way. This is consistent with the histories of adult women, where 93% had camouflaged [1]. Part-way into the study period, the clinical question as to whether school observations had helped with the diagnosis was added and there was a 100% response from 69 families with a school recognition failure of 72%. Later, it was realised that this was a far more important question for diagnosing girls since the 88% is the rate of the behaviour and the 72% is the rate of failure to recognise the behaviour at school. Social camouflaging at school is not just hiding ASD from the class teacher but also from the special needs teacher and the guidance officer and 72% of families described school observations as having not been of use in the initial diagnosis. The number of responses was not extended because by not recognising the importance of this variable when getting the histories, the responses were not unconsciously biased and affected the accuracy. This value would, of course, include other subsets of recognition bias, as well as camouflaging.

The two settings where children are observed closely over time are at school and at home, and individual families without a known child with ASD are unlikely to be as skilled in recognition as school officers charged with dealing with these sorts of problems. If we then assume those who are not recognised at all camouflage in general better than those who are recognised by an agency other than school, then the proportion of those diagnosed who are missed at school of 72% sets a lower bound on the proportion of total females with ASD not recognised. Then, the proportion of those not recognised will be 72 + δ%, which is independent corroboration of the recognition bias using information about female behaviour only.

### 2.14. Aftermath

In early 2020, there were two major changes to ASD diagnosis in Queensland. State funding of school assistance no longer required medical sign off and the National Disability Insurance Agency did not require medical sign-off for non-medical assistance for ASD. In addition, medical services were no longer face to face because of the pandemic and it was unclear what the overall COVID effect would be. We will examine the effect on the data to see whether terminating the study at that point was justified.

The only remaining absolute requirement for medical input was the prescription of medication for the comorbid conditions of ADHD; sleep problems; anxiety; and angry, irritable or violent behaviour. This would inevitably skew the referrals to boys and to the singleton girls who were in the category already recognised due basically to behaving like the boys. Siblings would span the range of behavioural severity, but it was thought likely that in families with prior experience of managing ASD, the better-behaved girls would possibly have their assessment held back due to the COVID-19 pandemic and then be less likely to need medical referral anyway. The result would be that the referred population would be skewed with fewer girls, in particular, sibling girls, and the MFOR would be factitiously raised. The final dataset of 2246 children up to 02 July 2021 is presented below in Table 5.

We first examined the entire internal category and compared the internal singletons and siblings. The change in sibling singleton ratio with time is shown in Table 6.

These results show that, overall, both fewer girls and fewer siblings were being diagnosed. Within the sibling category, the MFOR had risen. The MFOR up to 17 April 2020 was 0.7493. The MFOR from 18 April 2020 to 02 July 2021 was 1.302. A χ^2^ for the comparison of proportions gave *p* = 0.0090; therefore, as predicted, female siblings were clearly not being referred for diagnosis. The proportion of siblings referred for diagnosis having seen an AHP first prior to 17 April 2020 was 38.2%. The proportion after this date was 26.2%. The *p*-value using χ^2^ was 0.0085, supporting the hypothesis that they would not be referred unless a doctor was needed. The ratios for siblings who saw either an AHP or myself first both rose, suggesting that fewer girl siblings were presenting to anyone for diagnosis. This was most likely due to the COVID-19 pandemic. Overall there was a major difference in referral patterns from early 2020, justifying termination of the study. It also demonstrated the serious distortion of results when the sample population was not truly representative.

### 2.15. Summation

This study used a total of four methods to determine the recognition and/or ascertainment biases. The method using only the clinical database arrived at two possible sets of values for each. A value of the ascertainment bias was derived from the combination of the data-derived MFOR and a recent estimate of female prevalence in the United States. Another value was derived using entirely independent published data on the relationship between ASD and borderline personality disorder using Bayes’ Theorem combined with the US female prevalence. An estimate of recognition bias was made from recognition failure at school using the clinical histories of my patients. The mean of each set of estimates was then found. This ensemble averaging was used for estimating the tracks of tropical cyclones [12]. The values were not weighted since there was no evidence that one estimate was better than another. The final working values were rounded to the nearest whole percentage (Table 7) and the final variables are tabulated. The characteristics of the biases meant that estimates of recognition bias of 4 and ascertainment bias of 5 are sufficiently accurate for clinical and planning purposes. 

The original estimate of diagnosis bias Ba/Br assumed my diagnosis of girls was unbiased, with a value of 1.265. The final working estimate after rounding Ba and Br from all the methods was 1.25. We can assess whether my patient-derived value of 1.265 is externally valid by calculating the value from the estimates of Br and Ba derived separately from my diagnostic practice. The value of Ba based on the Bayesian BPD value and the US female prevalence was 100/(100-79). The most likely estimates of Br from the school non-recognition, which were based on parental experience prior to referral with δ = 1 or 2, were 100/(100–73) and 100/(100–74), and δ > 2 would make Bd insignificant. These estimates gave a likely range of Bd of 1.238 to 1.286, with a mean of 1.262 compared with the study diagnosis value of 1.265. 

## 3. Discussion

### 3.1. Outcome of the Biases and Future Directions

Currently, while diagnosis bias is important for individual patients, the bigger problem for all girls with ASD is the lack of initial recognition. If diagnosis were to improve so that half those currently missed were found, there would, on a population basis, be only a 2.5% improvement. Due to the hyperbolic nature of the bias relationship, a significant advance in recognition would rapidly lead to diagnosis bias becoming quantitatively more important as the values move to the left on the hyperbola. If the recognition bias were halved, those missed on diagnosis would double to 10% of the ASD population without any change in diagnostic practice. The two problems need to be tackled in tandem to remedy what is currently a very serious failure of clinical intervention.

There is emerging qualitative research from ASD women on their camouflaging practices [1]. This needs to be combined with similar information from the caregivers: frontline health workers who first encounter the problem, teachers, childcare workers and, above all, families, including those who have ASD children and those who do not. There will be other reasons for missing girls, but camouflaging is likely to be an upstream causal factor in all of them. This information will provide the context for designing programs to better educate all those potentially involved in finding girls with ASD.

### 3.2. Hypothesis

Is there evidence for a reason why girls might outnumber boys? The aetiology of ASD is largely but not entirely genetic [13] and there may well be a cultural component. Section D of the DSM-5 criteria [4] (p. 57) requires the entity to cause clinically significant impairment. There is abundant qualitative evidence of the distress girls with ASD suffer growing up [1,11,14]. As a clinician, it is clear that the central problem of ASD is poor reciprocal communication and the social communication expectations for girls are higher than for boys. Studies showing gendered genetic differences have a problem. For any study to demonstrate a true quantitative gender difference in any characteristic, such as gene distribution, the male and female populations studied, which are the denominators of the proportions of the characteristic compared, must not have different degrees of ascertainment bias. Unless the gender samples are truly representative, the proportions of the characteristic cannot be compared [14]. There are at least 102 genes to distribute [15], suggesting the most economical theoretical genetic contribution to the MFOR is 1:1. The cultural pressure leading to a diagnosable disorder will then tip the balance to females. Camouflaging appears to begin very early and there must be environmental factors with networked causal pathways for camouflaging and other factors, which might include the fact that girls are
“Girly” in their interests and the subtleties of approach to their interests are missed;Less likely to externalise and more likely to be anxious;Inherently more socially aware;Better in their apparent language skills, at least initially;Better at masking during tests;Disadvantaged by tests tailored more for boys.


Increased focus on girls will elucidate these bias subtypes, their prevalences and interactions, but here I frame no hypotheses. 

### 3.3. Why Does This Matter?

From these results, it appears that for every 1000 women, about 60 have clinical ASD. By 18 years of age, 12 have been diagnosed and 48 have not. If this is true of ASD in women, then it is a significant upstream factor in female mental health. Childhood and adolescence are very difficult for girls with undiagnosed ASD and constant psychic trauma is inevitable, including vulnerability to sexual exploitation [11]. My clinic does not diagnose adults, but from my experience with the histories of diagnosed or probably affected mothers, which are common due to the high heritability [13], anxiety is very common in ASD and adult women describe relentless mental trauma from a young age with no cause found, or worse, a whole gamut of incorrect or incomplete causes. It appears that women with ASD get an alphabet soup of diagnoses, including borderline personality disorder, eating disorders, bipolar disorder, schizoaffective disorder, schizophrenia, post-traumatic stress disorder, sensory processing disorder, intermittent explosive disorder and adult ADHD, as well as the varieties of anxiety, agoraphobia, panic disorder and depression, serially and together. A disturbing number of mothers seek my advice on a pathway to diagnosis. ASD is not a mental illness and an individual may have one or more of these conditions as comorbidities, but without the upstream causal factor of ASD identified, she will never fully get to grips with her condition and gain psychic relief by understanding herself.

### 3.4. Autistic Spectrum Disorder and Autistic Spectrum Condition (ASC)

There is a view that the term ASD is stigmatising and ASC will serve for both the strengths and difficulties of those on the spectrum [1]. Ambiguity is not helpful in a clinical discussion. Each term has value in context. Clinicians only make a diagnosis if there is a disorder. Many of those with features of the spectrum adapt without clinical intervention. An ASC is not a disease to “cure” or a disability to treat. For those whose minds are disordered, the diagnosis is the fork in the road. They then have the opportunity through self-understanding and therapeutic intervention to transition to an ASC. This is the satisfactory endpoint. The other tine of the fork leads to continuing disorder or descends into mental illness. From my observations, the perceived stigma of a diagnosis is definitely a feature of both recognition and diagnosis bias. Clear differentiation between ASD and ASC and seeing diagnosis as the key to enabling the transition will reduce bias, in particular for girls. 

## 4. Conclusions

From a practical perspective, the results of this study need only be sufficiently precise for informed decisionmaking regarding diagnosis, management and service development. This is provided by

# Three variables: the male/female odds ratio, the recognition bias and the diagnosis bias, which are described by three numbers: 3/4, 4 and 5/4.

# Three rules: as biases occur, their values multiply, the proportion of girls found is the reciprocal of the bias and, *above all*, they attend to the mother’s history.

## Figures and Tables

**Figure 1 children-09-00272-f001:**
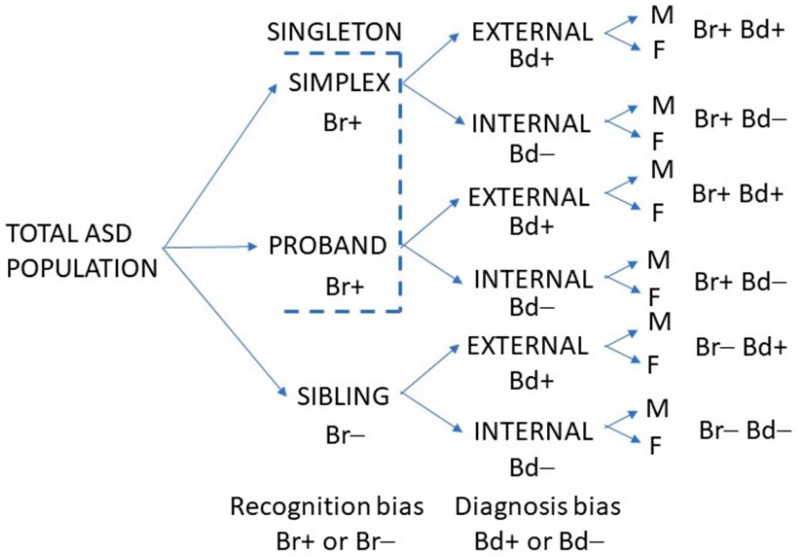
Categorising the ASD Study Population.

**Figure 2 children-09-00272-f002:**
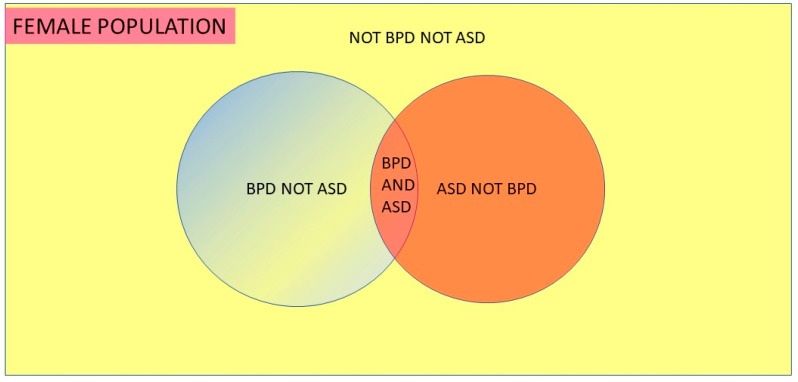
Venn Diagram of Bayes’ Theorem.

**Figure 3 children-09-00272-f003:**
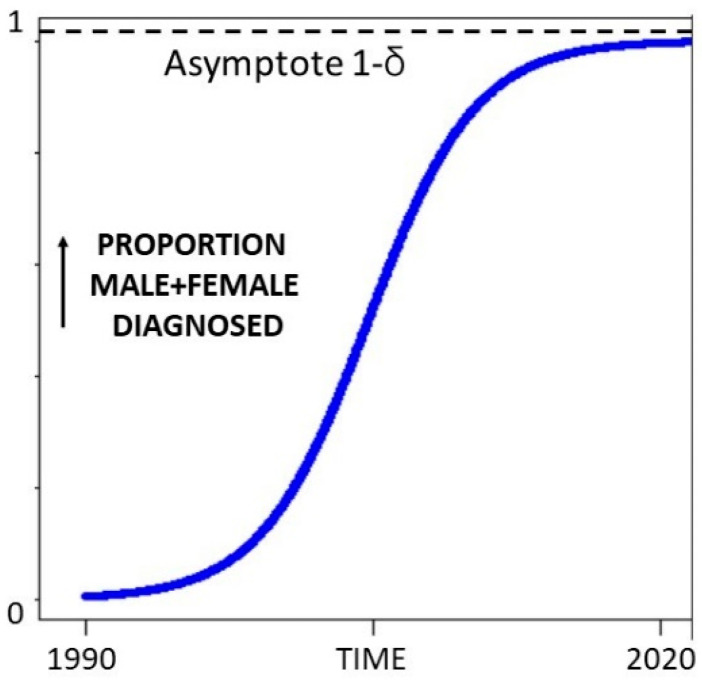
Learning curve of ASD over time.

**Figure 4 children-09-00272-f004:**
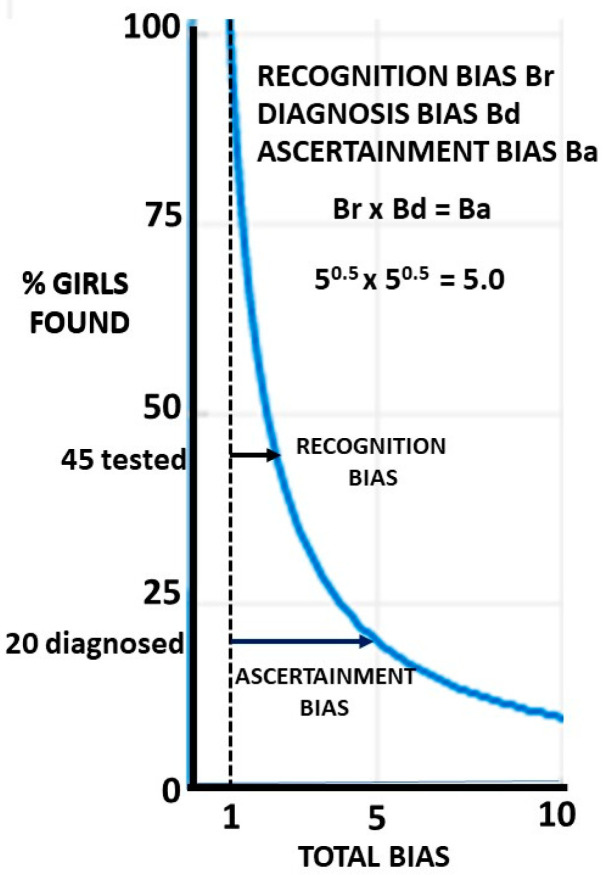
The effect of equal biases on the proportion of girls found.

**Figure 5 children-09-00272-f005:**
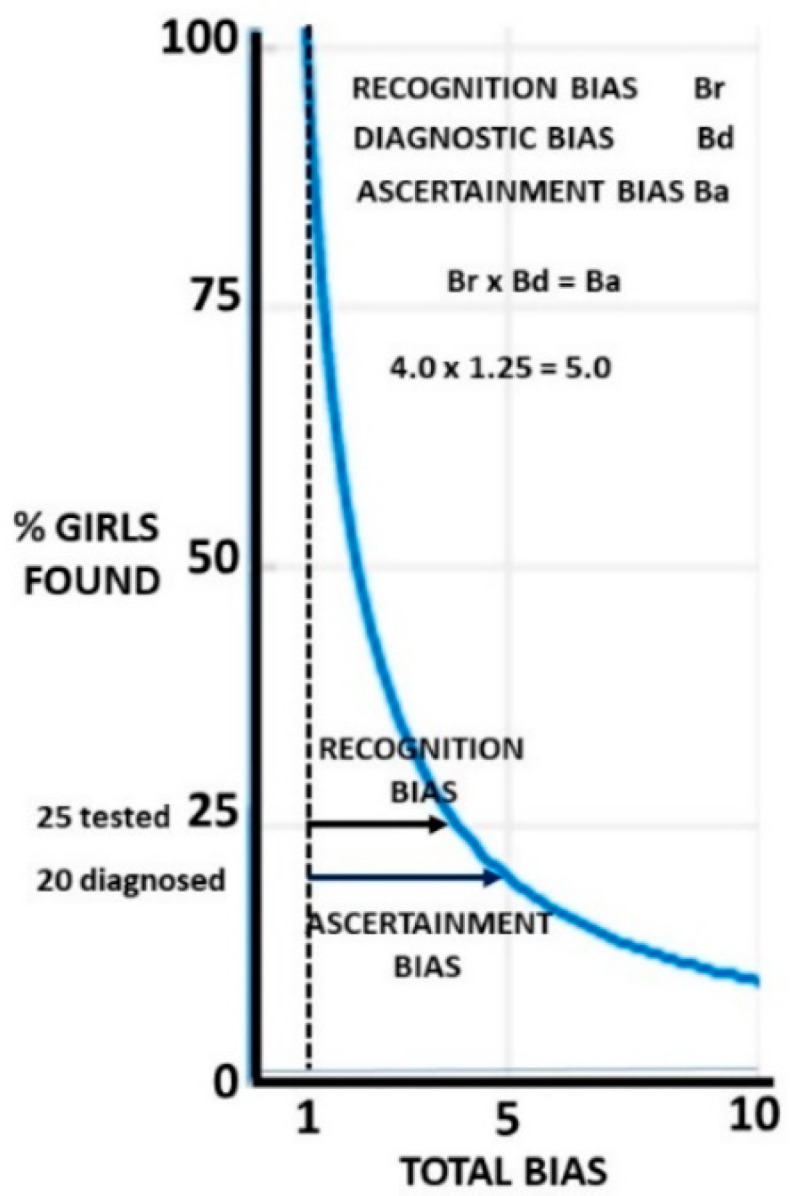
The effect of the final biases on the proportion of girls found.

**Table 1 children-09-00272-t001:** Data for sibling categories.

Diagnosis	Male	Female	uMFOR	MFOR
External	44	39	1.128	1.069
Internal	200	253	0.7905	0.7493
Doctor alone	95	127	0.748	0.709
Doctor + AHP	105	126	0.883	0.790
AHP 1st	75	98	0.765	0.725
Doctor 1st	30	28	1.071	1.016

**Table 2 children-09-00272-t002:** Percentages of categorised females: by age {columns} and diagnostic pathway (rows).

Category	<3 Years	3 + 4 Years	5 + 6 Years
Doctor alone	{19} (13.7)	{48} (34.5)	{72} (51.2)
Doctor first	{22} (66.7)	{11} (33.3)	0
AHP first	{59} (46.1)	{41} (32.0)	{28} (21.9)

**Table 3 children-09-00272-t003:** Notation scheme for variables.

Entity	Subscript	Subscript
B—bias	a—ascertainment	m—male
D—population	b—sibling	o—no bias
N—number	d—diagnosis	p—proband
P—proportion	f—female	r—recognition
R—odds ratio	i—reciprocal	u—unadjusted

**Table 4 children-09-00272-t004:** Complete dataset of 1711 children for the calculations.

Category	Diagnosis	Male	Female	uMFOR	MFOR
Simplex	External	142	57	2.491	2.361
	Internal	376	193	1.948	1.847
Multiplex proband	External	119	48	2.479	2.350
	Internal	171	69	2.478	2.349
Sibling	External	44	39	1.128	1.069
	Internal	200	253	0.791	0.7493

**Table 5 children-09-00272-t005:** Final dataset of 2246 children.

Category	Diagnosis	Male	Female	uMFOR	MFOR
Simplex	External	192	76	2.526	2.395
	Internal	512	251	2.040	1.933
Multiplex proband	External	136	59	2.305	2.185
	Internal	224	101	2.218	2.102
Sibling	External	49	48	1.021	0.968
	Internal	282	316	0.8924	0.8459
	Doctor alone	150	169	0.888	0.841
	Doctor + AHP	132	147	0.898	0.851

**Table 6 children-09-00272-t006:** Internal group.

Time of Data Collection	Total uMFOR	Sibling Singleton Ratio
Up to 17 April 2020	1.450	0.560
Up to 02 July 2021	1.524	0.550
18 April 2020 to 02 July 2021	1.771	0.520

**Table 7 children-09-00272-t007:** Pathways to finding girls missed.

Information Source	Not Recognised (%)	Not Ascertained (%)
Study + US prevalence		77
Borderline personality		79
School recognition	72 + δ	
Study alone	73.1, 79.1	78.8, 83.4
Mean, working value and bias	74.7 + δ/3, 75, 4	79.6, 80, 5

## Data Availability

All the data for key results, methods devised and equations derived are in the text, both for checking calculations and reproducing the study. Data assisting the patient categorisation, e.g., gender, chronological age and age at diagnosis, has not been deidentified in my database and is not presented. It is not essential to the argument and would be unique to any study replication.

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
