# Peer review of "Finding the True Number of Females with Autistic Spectrum Disorder by Estimating the Biases in Initial Recognition and Clinical Diagnosis"

_children, 2022, doi:10.3390/children9020272_

Round 1

Reviewer 1 Report

In this interesting study, the author revisits the generally admited ratio of males to females with ASD (4:1) by taking into account biases leading to potential falses negatives in females. The author starts by identifying and quantifying the biases, and then calculates the proportion of females with ASD that are missed in. Finally, the author proposes a mathematical model to compare his results with current published data. The author concludes that the true male to female ratio is 3:4, suggesting that 80% of females with ASD are false negatives.

Early diagnosis drastically ameliorate the prognosis of ASD patients. This study is of timely importance and has high clinical value, as it should help designing new programs that neutralize the identified biases (e.g., camouflaging).

I have only minor comments/suggestions:

1) I suggest using a shorter the title that directly reflects the main finding presented in the manuscript. Using "and" twice should be avoided.

2) I suggest discussing the possibility of applying the strategy used by the author in ASD patients to other disorders where biases could affect the diagnosis, if it is likely to be the case. That would attract the attention of a wider readership, and help improving the diagnosis of other disorders and mental diseases.

3) Please avoid using "sibs" instead of "siblings".

4) I won't provide details about formatting, but the text will have to be re-checked. I found many extra spaces/tabs and typos.

The overall aim of this well-written manuscript is laudable and the results reported are of timely importance.

Reviewer 2 Report

The current study is sound, timely and interesting. The statistical method applied is noteworthy and well explained, maybe with too many details that could make the reader lose his/her interest. The paper is globally well written, although maybe too much of the text is dedicated to the presentation of (correct) statistical arguments, and only the final sections specifically deal with the practical relevance of study findings, with clinical remarks and with the possibility of building on study results. Maybe more could be said about the literature evidence on the female autism clinical phenotype, or about sex/gender differences in the age of diagnosis/diagnostic assessment methods. The reference list could be expanded, for example adding some specific studies assessing the gender ratio in clinical samples with autism spectrum disorders. A section dedicated to study limitations - and mentioning other possible reasons for preventing females being diagnosed with autism - could be added.
